# DINO is Also a Semantic Guider: Exploiting Class-aware Affinity for Weakly Supervised Semantic Segmentation

Submission Id: 853

## ABSTRACT

Weakly supervised semantic segmentation (WSSS) using image-level labels is a challenging task, with relying on Class Activation Map (CAM) to derive segmentation supervision. Although many efficient single-stage solutions have been proposed, their performance is hindered by the inherent ambiguity of CAM. This paper introduces a new approach, dubbed **ECA**, to **E**xploit the self-supervised Vision Transformer, DINO, inducing the **C**lass-aware semantic **A**ffinity to overcome this limitation. Specifically, we introduce a Semantic Affinity Exploitation module (SAE). It establishes the class-agnostic affinity graph through the self-attention of DINO. Using the highly activated patches on CAMs as "seeds", we propagate them across the affinity graph and yield the Class-aware Affinity Region Map (CARM) as supplementary semantic guidance. Moreover, the selection of reliable "seeds" is crucial to the CARM generation. Inspired by the observed CAM inconsistency between the global and local views, we develop a CAM Correspondence Enhancement module (CCE) to encourage dense local-to-global CAM correspondences, advancing high-fidelity CAM for seed selection in SAE. Our experimental results demonstrate that ECA effectively improves the model's object pattern understanding. Remarkably, it outperforms state-of-the-art alternatives on the PASCAL VOC 2012 and MS COCO 2014 datasets, achieving 90.1% upper bound performance compared to its fully supervised counterpart. [1]

## CCS CONCEPTS

• **Computing methodologies** → **Computer vision**.

## KEYWORDS

Weakly Supervised, Image Segmentation, Semantic Affinity

## 1 INTRODUCTION

Semantic segmentation is a fundamental computer vision task that assigns a class for every pixel in an image. Despite the substantial progress in deep-learning-based segmentation models, their performance relies on large-scale pixel-level annotations. Weakly supervised semantic segmentation (WSSS) aims to address this shortcoming by training segmentation models using weak and cost-effective labels, such as sparse points [4], bounding boxes [10, 21], and image-level labels [24, 29, 30]. Among these WSSS tasks, the most rewarding yet challenging one is segmentation using only image-level labels because of its minimum information capacity. This paper also falls within the domain of training a segmentation model using only image-level labels.

Prevalent WSSS with image-level labels commonly follow a multi-stage pipeline: 1) generating pixel pseudo-labels from Class

---
[1]Code will be available after acceptance.

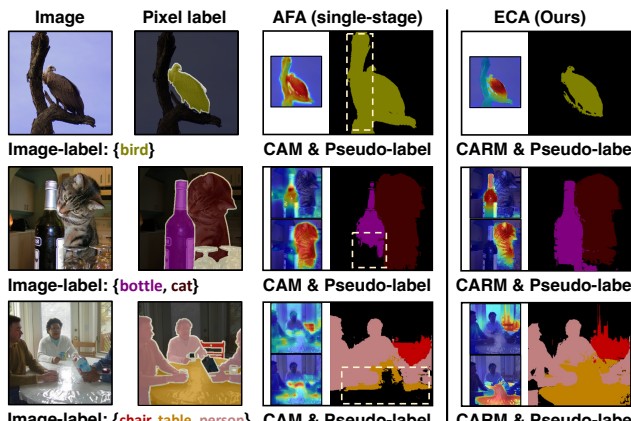

**Figure 1: The produced pseudo-labels in single-stage WSSS. The CAMs are from the prior art, AFA [24]. The proposed ECA's Class-aware Affinity Region Maps (CARM) from DINO can produce more accurate pseudo-labels.**

Activation Maps (CAM) [38]; 2) training an affinity refinement network [1, 2] with dense Conditional Random Field (CRF) to refine the pseudo-labels; 3) training segmentation networks [6, 31] using the refined pseudo-labels. This pipeline necessitates training multiple models, thus complicating WSSS and diminishing the training efficiency. To address this issue, some recent works attempted to devise efficient single-stage solutions based on end-to-end Convolutional Neural Network (CNN) [3, 34] and Vision Transformer (ViT) architectures [24, 25].

However, compared to the pseudo-labels refined by training a dedicated refinement network in the multi-stage WSSS, those directly generated by the CAMs in single-stage methods exhibit inferior quality. This dramatically impairs segmentation supervision, resulting in subpar segmentation outcomes. **One *critical reason* is the inevitable ambiguity of CAM that stems from the gap between the classification and localization tasks.** As illustrated in Figure 1, CAM tends to cover non-target regions (*i.e.*, over-activations) or only the most discriminative regions (*i.e.*, under-activations). Despite some post-processing modules integrated in recent single-stage works [24, 33], which introduce local RGB and position information to alleviate this problem, they still struggle to produce high-quality pseudo-labels. We can clearly observe that the flaws existing in CAMs still remain in the produced final pseudo-labels, indicating the inadequacy of leveraging RGB and position information to calibrate them.

Recently, the self-supervised Vision Transformer (ViT) model [11], DINO [5, 22], has shown the capability of naturally modeling pairwise affinity between two image patches without using any annotations. This property allows for the grouping of patches with

similar semantic properties. Although these grouped components are "class-agnostic" and independent of predefined categories (*e.g.*, *head* and *body* rather than *person* and *car*), the coverage of each component is semantically precise and complete, and the patches within the same component have strong affinity. This merit can reveal the regions that CAM is difficult to cover accurately. *Conversely*, CAM cannot produce accurate object coverage, but it can provide class and localization information. Based on this insight, **our *objective* is to leverage the CAM's class and localization priors to induce the task-required class-aware affinity from DINO as the semantic guidance to overcome the above issue.**

This paper proposes a new framework, ECA, to improve the performance of single-stage WSSS through the exploitation of DINO. Specifically, we propose a DINO-guided Semantic Affinity Exploitation module (SAE) to generate Class-aware Affinity Region Map (CARM), offering the complementary semantic guidance of CAM. For each image, SAE comprises two steps motivated by the vanilla "Seed and Expansion" pipeline. The first step, "*selection of initial seeds*", involves selecting top $k\%$ activations on the CAM to serve as the initial seeds. The second step, "*generation of class affinity region map*", entails constructing a class-agnostic affinity graph by measuring pairwise similarities among the patch keys derived from DINO [5]. The seeds are then propagated on the affinity graph, yielding the CARM of each labeled class for supervision. Empirical evidence shows that the generated CARM effectively mitigates the detrimental impact of CAM ambiguity on segmentation supervision, leading to notable performance improvement. Furthermore, to select better initial seeds, we introduce the CAM Correspondence Enhancement (CCE). Inspired by the inconsistency of CAM under different views [29], CCE encourages local-to-global correspondences of CAM, providing high-fidelity CAM for the seed selection. Overall, our contributions are summarized as follows:

- We propose a Semantic Affinity Exploitation module (SAE) to fully exploit the potential of semantic affinity existing in DINO. It produces powerful complementary semantic guidance, *i.e.*, Class-aware Affinity Region Map (CARM), to boost the single-stage WSSS performance.

- We propose a CAM Correspondence Enhancement module (CCE) to promote the dense local-to-global correspondences of CAM. It is proved that CCE promotes the object activation completeness and the robustness of CAM, which benefits the seed selection in SAE module.

- Building upon SAE and CCE, the proposed ECA can significantly outperform single-stage WSSS competitors on the PASCAL VOC 2012 and MS COCO 2014 dataset, and achieve comparable performance with multi-stage methods. To the best of our knowledge, ECA is the first work that derives semantic affinity of DINO for WSSS.

## 2 RELATED WORK

### 2.1 Single-stage WSSS

Compared with the multi-stage WSSS pipeline, single-stage solutions learn a segmentation decoder using the generated pseudo-labels in an end-to-end manner. Due to the lack of satisfactory pseudo-labels, the performance of single-stage WSSS models lags behind multi-stage models. To address this limitation, many studies focused on optimizing CAM pseudo-labels [3, 24, 33]. For instance, RRM [35] employed Conditional Random Field (CRF) to produce the refined label as supervision for segmentation. AFA [24] introduced a Pixel-Adaptive Refinement (PAR) module to refine the CAM pseudo-labels effectively. Additionally, it explores the use of Transformer with an affinity learning loss to enforce the model to learn object-related attention, resulting in better pseudo-labels for segmentation. TSCD [33] introduced a CAM-based self-distillation task and proposes a Variation-aware Refine Module to achieve better segmentation. However, we observe that these strategies mainly depend on the RGB and position priors of pixels to refine CAM pseudo-labels, making them difficult to calibrate low-quality pseudo-labels that are severely impaired by CAM ambiguity. *In this work*, we use CAM to induce the class-aware affinity map from the self-supervised ViT, DINO, as the complementary guidance. This significantly enhances the performance of single-stage WSSS.

### 2.2 Self-supervised Foundation Model

Self-supervised learning aims to learn visual representations from unlabeled images by solving pretext tasks [15]. In this manner, their pretrained foundation models enable to outperform their supervised counterparts when transferring to downstream tasks. DINO [5], a self-supervised ViT model, has been observed that it automatically emerges underlying class-agnostic semantic properties, which is not explicitly present in any supervised ViT or CNN [5, 40]. Leveraging the unique property of DINO, several studies proposed to detect a single salient object from each image using a graph constructed with DINO's patch tokens [26, 28]. *In contrast*, WSSS aims to segment single or multiple objects with specific semantics for each image. The proposed ECA integrates the CAM's class and position priors to induce each labeled class's class-aware affinity region via the constructed class-agnostic affinity graph from DINO. This perspective has not been studied in previous WSSS works.

## 3 PROPOSED METHOD

This section begins with a brief review of generating CAM from a Transformer backbone. Then, Semantic Affinity Exploitation module (SAE) is presented to exploit the class-aware affinity region map (CARM) for each labeled class. Next, CAM Correspondence Enhancement (CCE) is introduced to align the CAM activations between the global and local views. Lastly, we present the total training objective of ECA. The overview of the proposed ECA framework is illustrated in Figure 2.

### 3.1 Class Activation Map in Transformers

Initially, CAM is introduced for CNN to localize the discriminative regions for specific classes [38]. Because of its effectiveness and simplicity, CAM has been prevalently employed to generate the initial pseudo-labels for WSSS. For Transformer-based models, CAM is generated in a similar manner as that on CNN. Given an image, its patch tokens are treated as a feature map $\mathbf{F} \in \mathbb{R}^{D \times HW}$, where HW denotes the spatial dimension, *i.e.*, the number of patches N, and D denotes the channel dimension. Subsequently, it is sent to a Global Max-Pooling layer with a classification layer to retrieve the classification score. In the above process, given C classes, the

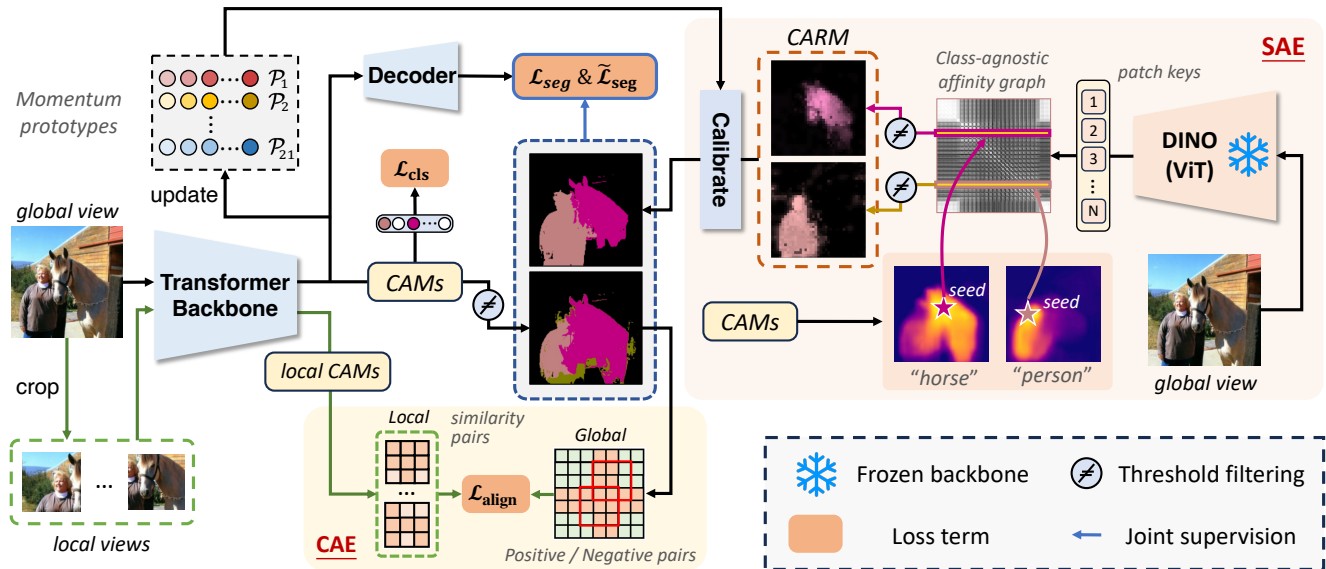

**Figure 2: The framework overview of ECA. ECA first generates CAMs from a ViT-variant backbone, Mix Transformer (MiT) [31]. In Semantic Affinity Exploitation (SAE), the input image is fed to the frozen DINO model to generate Class-agnostic affinity graph. Then the seeds from CAMs are used to induce Class-aware Affinity Region Maps (CARM). CARMs are subsequently calibrated by using momentum prototypes. In CAM Correspondence Enhancement (CCE), we use global CAM pseudo-labels to construct dense correlation pairs to supervise the correspondence between global and local patch CAMs. For the segment decoder, it is jointly supervised by CAM and CARM pseudo-labels.**

CAM of class $c$, denoted as $\mathbf{M}_c \in \mathbb{R}^{HW}$, is generated by weighting and summing the feature map $\mathbf{F}$ using the classification weights $\mathbf{W} \in \mathbb{R}^{D \times C}$:

$$\mathbf{M}_c = \mathsf{ReLU}\left(\sum_{i=1}^{D} \mathbf{W}_{i,c} \cdot \mathbf{F}_i\right), \tag{1}$$

where ReLU function is used to filter out negative activations. Then, min-max normalization is applied to rescale $\mathbf{M}_c$ to $[0, 1]$. For the background (class 0) CAM, denoted as $\mathbf{M}_0$, the value of patch at position $i$ is defined as:

$$\mathbf{M}_0(i) = 1 - \max_{1 \leq c \leq C} \mathbf{M}_c(i), \tag{2}$$

where $\mathbf{M}_c(i)$ is the CAM activation value of class $c$ at patch $i$. Note that the background CAM is only adopted in Semantic Affinity Exploitation (Equation 3). During the pseudo-label generation stage, a hard background threshold $\alpha$ is set to separate the foreground objects and background.

### 3.2 Semantic Affinity Exploitation

**Selection of Initial CAM Seeds.** The CAM can provide the class and localization priors to induce the class-aware regions from DINO. Thereinto, the value of CAM can be regarded as confidence scores to select reliable initial seeds of its labeled class. However, some regions are often activated in multiple classes and may erroneously dominate the high-confident activations in multiple classes, producing incorrect seeds for subsequent processes. To this end, for each image's CAM, *i.e.*, $\mathbf{M} \in \mathbb{R}^{(C+1) \times HW}$, we deploy a mask $\mathbf{I}_{\mathrm{cam}} = \mathrm{argmax}_c(\mathbf{M})$ to ensure that the position of each image patch is exclusively activated in one class during seed selection. Then,

the initial CAM seeds of its labeled class $c$ are those patches within the top $k\%$ of nonzero activations:

$$Q_c = \mathrm{Top}_k \left(\mathbf{M}_c \odot \mathbb{I}\left(\mathbf{I}_{\mathrm{cam}} = c\right)\right), \tag{3}$$

where $Q_c$ represents the collection that records the position of class $c$'s initial seeds, and $\odot$ denotes the Hadamard product. In the above selection process, hyperparameter $k$ controls the percent of patches considered, where a smaller $k$ indicates higher confidence for selected initial seeds.

**Construction of Class-agnostic Affinity Graph.** DINO [5] demonstrates that its self-supervised ViT can learn a perceptual grouping of image patches. Therefore, we leverage this merit to model the pair-wise affinity between any two image patches, which is noticeable when using the *keys* from its self-attention heads [26, 27]. For each image, given the *keys* of N patches from the last self-attention block of the frozen DINO, we construct an undirected class-agnostic affinity graph $\mathcal{G}_{\mathrm{aff}} \in \mathbb{R}^{N \times N}$. The affinity between patch $i$ and patch $j$, denoted as $\mathcal{G}_{\mathrm{aff}}(i, j)$, can be calculated by:

$$\mathcal{G}_{\mathrm{aff}}(i, j) = \frac{\mathbf{k}_i \cdot \mathbf{k}_j^{\mathrm{T}}}{\|\mathbf{k}_i\|_2 \times \|\mathbf{k}_j\|_2}, \quad \{\mathbf{k}_i, \mathbf{k}_j\} \in \mathbb{R}^{1 \times D}, \tag{4}$$

where $\mathbf{k}_i$ and $\mathbf{k}_j$ are the key of the patch at spatial position $i, j \in [0, N-1]$. A higher $\mathcal{G}_{\mathrm{aff}}(i, j)$ indicates stronger affinity between patch $i$ and patch $j$, implying that these two patches tend to belong to the same labeled class.

Moreover, since the class token interacts with patches to represent the image, the attention between class token and patches

can reveal valuable foreground object cues that facilitate the filtering of background patches. Thus, similarly to the construction of $\mathcal{G}_{\text{aff}}$, we establish the foreground-cue map for each image, denoted as $\mathbf{I}_{\text{fg}} \in \mathbb{R}^{1 \times N}$, by measuring the similarity between the *query* of the class token and the *keys* of N patches. For the patch at spatial position $i \in [0, N-1]$, its value of $\mathbf{I}_{\text{fg}}(i)$ can be calculated as:

$$\mathbf{I}_{\text{fg}}(i) = \frac{\boldsymbol{q} \cdot \boldsymbol{k}_i^{\text{T}}}{\|\boldsymbol{q}\|_2 \times \|\boldsymbol{k}_i\|_2}, \quad \{\boldsymbol{q}, \boldsymbol{k}_i\} \in \mathbb{R}^{1 \times D}, \quad (5)$$

where $\boldsymbol{q}$ denotes the query of the class token, while $\boldsymbol{k}_i$ represents the key of the patch $i$. A higher value of $\mathbf{I}_{\text{fg}}(i)$ indicates a stronger affinity between the class token and patch $i$, suggesting that patch $i$ is more likely to be part of the foreground.

*Note that* the DINO model used in the above process is always kept frozen, serving only for inference purposes. This ensures that the DINO model does not affect too much training efficiency. We exclude the Multi-head Self-attention mechanism in the above construction of $\mathcal{G}_{\text{aff}}$ and $\mathbf{I}_{\text{fg}}$ (Equation 4 and 5) for brevity. In actual implementation, the *query* and *keys* from each self-attention head are concatenated together to derive $\mathcal{G}_{\text{aff}}$ and $\mathbf{I}_{\text{fg}}$.

**Generation of Class-aware Affinity Region Map.** For each image, its initial CAM seeds are used to induce $\mathcal{G}_{\text{aff}}$ to generate its corresponding Class-aware Affinity Region Map (CARM). Assuming that the image is labeled as $c$, the seeds from its $Q_c$ first propagate along the rows of $\mathcal{G}_{\text{aff}}$, constructing the binary adjacency vectors $\mathbf{A}$ based on a threshold $\gamma$. Formally, for the seed at position $i \in Q_c$, the above process can be formulated as:

$$\mathbf{A}(i) = (\mathcal{G}_{\text{aff}}(i, :) > \gamma), \quad \text{where } \mathbf{A}(i) \in \mathbb{R}^{1 \times N}. \quad (6)$$

Subsequently, the binary adjacency vectors $\mathbf{A} \in \mathbb{R}^{|Q_c| \times N}$ are aggregated with the foreground-cue map $\mathbf{I}_{\text{fg}}$ to construct the original CARM of class $c$, denoted as $\mathcal{M}_c$:

$$\mathcal{M}_c = \begin{cases} \sum_{i \in Q_c} \mathbf{A}(i) \odot \mathbf{I}_{\text{fg}}, & \text{if } c \in [1, C-1] \\ \sum_{i \in Q_0} \mathbf{A}(i), & \text{if } c = 0 \end{cases}, \quad (7)$$

where the value of $\mathcal{M}_c \in \mathbb{R}^{1 \times N}$ intuitively represents the degree of each patch from a graph perspective. A patch with a higher degree in $\mathcal{M}_c$ suggests that it is more likely to belong to class $c$. Finally, we reshape $\mathcal{M}_c$ to the size of H×W and applied min-max normalization to rescale its values.

**Calibration of Class-aware Affinity Region Map.** After generating CARM $\mathcal{M}$ for each image, it can be utilized as the semantic guidance for WSSS, following a similar manner as CAM. However, our early experiment indicates that CARM is susceptible to erroneous initial CAM seed selection owing to the over-activation issue, *i.e.*, identifying the background patch as that of target objects. This situation will lead to imprecise propagation on the class-agnostic affinity graph and produce noisy CARM, which in turn degrades the final segmentation performance.

To tackle this issue, we introduce the momentum prototype for each class to calibrate CARM. The *motivation* is that the momentum prototypes, updated by the tokens across the entire datasets, can serve as more robust class-representative tokens compared to the tokens from image instances. Consequently, we compute the similarity between image tokens and the prototypes to suppress

unreliable CARM regions. Specifically, the momentum prototypes $\mathbf{P}$ are updated by the tokens corresponding to the seeds from $Q$. For each class $c$, its prototype $\mathbf{P}_c$ can be updated as:

$$\mathbf{P}_c \leftarrow \tau \cdot \mathbf{P}_c + (1 - \tau) \cdot \frac{1}{|Q_c|} \sum_{i \in Q_c} \mathbf{t}_i, \quad (8)$$

where $\mathbf{t}_i$ denotes the patch token of the selected seed at position $i$, $\tau$ is momentum for the prototype update, and $|\cdot|$ represents the collection cardinality. Then, given an image labeled as class $c$ with its feature map $\mathbf{F} \in \mathbb{R}^{D \times HW}$ and prototype $\mathbf{P}_c \in \mathbb{R}^{1 \times D}$, we calculate the similarity map $\mathbf{S}_c \in \mathbb{R}^{HW}$, where the value of the patch at position $i$, denoted as $\mathbf{S}_c(i)$, can be calculated as :

$$\mathbf{S}_c(i) = \max \left[ \frac{\mathbf{P}_c \cdot \mathbf{F}(:, i)}{\|\mathbf{P}_c\|_2 \times \|\mathbf{F}(:, i)\|_2}, 0 \right]. \quad (9)$$

Finally, the similarity map $\mathbf{S}_c$ is reshape to H × W and the image's calibrated CARM of class $c$ will be:

$$\mathcal{M}_c = \text{min-max}(\mathcal{M}_c \odot \mathbf{S}_c^2). \quad (10)$$

After calibration process, the CARM pseudo-label is produced as the complementary supervision of the segment decoder and the guidance of the semantic affinity relations in self-attention through affinity learning [24].

## 3.3 CAM Correspondence Enhancement

In the SAE module, due to the CAM ambiguity, the selected seeds sometimes fail to cover all components of target objects or be assigned to incorrect collections of $Q$. This phenomenon is exacerbated when the images have incomplete or local views, as noted by Wang et al. [29]. To alleviate the CAM inconsistency under the global and local views, CAM Correspondence Enhancement (CCE) is introduced to enhance the object activation completeness and the robustness of CAM for better seed selection. **The objective of CCE is to align the CAM activations between the global view and multiple local views through establishing dense correspondences.** Intuitively, if a pair of patches in the global view exhibits highly similar activations, the corresponding pairs of activations within the local views will be encouraged stronger similarity. Otherwise, the pair of activations in the local view will be encouraged to exhibit weaker similarity. The local views are generated by segmenting objects from the global view and applying re-augmentation to accentuate the discrepancy from the global view. To reduce the computational load, we utilize the CAM pseudo-label of the global view to establish the hard pairwise correlations (*i.e.*, the activation similarity between two patches assigned with the same pseudo-label is considered as 1, otherwise it is 0).

Following previous studies [24, 25], we begin by filtering uncertain regions of CAMs to produce CAM pseudo-labels. Next, the patches that share the same pixel pseudo-label in the global view are identified as positive pairs, whereas those with different labels are designated as negative pairs. As illustrated in Figure 3, the objective of CCE is two-fold: maximizing the similarity between patch activations belonging to positive pairs, and meanwhile, minimizing the similarity for those in negative pairs. This proposed

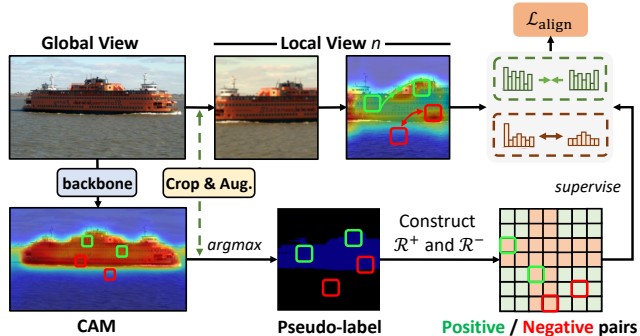

**Figure 3: Illustration of the CAM Correspondence Enhancement (CCE) module. Only one local view is presented for clear visualization.**

correspondence loss is formulated as follows:

$$\mathcal{L}_{\text{corr}} = \sum_{n=1}^{N} \Bigg[ \frac{1}{|\mathcal{R}_n^+|} \sum_{\{i,j\} \in \mathcal{R}_n^+} \left(1 - \mathcal{D}\left(\mathbf{M}_n\left(:, 1:i\right), \mathbf{M}_n\left(:, 1:j\right)\right)\right)$$
$$+ \frac{1}{|\mathcal{R}_n^-|} \sum_{\{i,j\} \in \mathcal{R}_n^-} \mathcal{D}\left(\mathbf{M}_n\left(:, 1:i\right), \mathbf{M}_n\left(:, 1:j\right)\right) \Bigg], \tag{11}$$

where $\mathbf{M}_n(:, 1:i) \in \mathbb{R}^C$ denotes the non-background CAM activation of patch $i$ at the local view $l$, and $(\mathcal{R}_l^+, \mathcal{R}_l^-)$ represent the collections of positive and negative pairs within each local view $l$, determined by the corresponding global CAM pseudo-label, respectively. Across all local views, from $l = 1$ to $N$, Jensen-Shannon divergence $\mathcal{D}$ is employed to measure the discrepancy between the activations of two patches as the training loss to promote the CAM's correspondence.

### 3.4 Network Training of ECA

As shown in Figure 2, except the classification loss $\mathcal{L}_{\text{cls}}$ and the joint segmentation loss of CAM and CARM pseudo-labels $\mathcal{L}_{\text{seg}}$ and $\widetilde{\mathcal{L}}_{\text{seg}}$, the proposed ECA also employ the auxiliary affinity learning loss $\mathcal{L}_{\text{aff}}$ [24] to transfer the knowledge from CARM to the semantic affinity relations in multi-head self-attention. Following the common practice, these CAM and CARM pseudo-labels are refined by the online post-processing module, *i.e.*, PAR [24]. The Multi-label Soft Margin Loss is adopted for $\mathcal{L}_{\text{cls}}$, and the Cross Entropy is adopted for both $\mathcal{L}_{\text{seg}}$ and $\widetilde{\mathcal{L}}_{\text{seg}}$. The optimization objective of ECA is the weighted sum of these loss terms:

$$\mathcal{L}_{\text{ECA}} = \mathcal{L}_{\text{cls}} + \lambda_1 \left(\mathcal{L}_{\text{seg}} + \widetilde{\mathcal{L}}_{\text{seg}}\right) + \lambda_2 \, \mathcal{L}_{\text{corr}} + \lambda_3 \, \mathcal{L}_{\text{aff}}, \tag{12}$$

where $\lambda_i, i \in \{1, 2, 3\}$ are the weights used to rescale the losses of different learning objectives.

## 4 EXPERIMENTS

### 4.1 Setup

**Datasets and evaluation metric.** We conduct our experiments on PASCAL VOC 2012 [13] and MS COCO 2014 datasets [19]. PASCAL VOC 2012 dataset comprises training, validation, and test sets with a total of 21 classes. Following the WSSS convention, we adopt

| Methods | BB. | VOC | | COCO |
|---|---|---|---|---|
| | | val | test | val |
| ***Multi-stage WSSS methods*** | | | | |
| RCA[†] [39] CVPR '22 | R101 | 72.2 | 72.8 | 36.8 |
| PPC[†] [12] CVPR '22 | R101 | 72.6 | 73.6 | - |
| ReCAM[†] [8] CVPR '22 | R101 | 68.4 | 68.2 | 45.0 |
| MCTFormer [32] CVPR '22 | WR38 | 71.9 | 71.6 | 42.0 |
| W-OoD [17] CVPR '22 | WR38 | 70.7 | 70.1 | - |
| ESOL [18] CVPR '22 | R101 | 69.9 | 69.3 | 42.6 |
| OCR [9] CVPR '23 | WR38 | 72.7 | 72.0 | 42.5 |
| ACR [16] CVPR '23 | R101 | 71.9 | 71.9 | 45.3 |
| ***Single-stage WSSS methods*** | | | | |
| RRM [35] AAAI '20 | WR38 | 62.6 | 62.9 | - |
| 1Stage [3] CVPR '20 | WR38 | 62.7 | 64.3 | - |
| AA&LR [36] ACM MM '21 | WR38 | 63.9 | 64.8 | - |
| SLRNet [23] IJCV '22 | WR38 | 67.2 | 67.6 | 35.0 |
| AFA [24] CVPR '22 | MiT-B1 | 66.0 | 66.3 | 38.9 |
| TSCD [33] AAAI '23 | MiT-B1 | 67.3 | 67.5 | 40.1 |
| ToCo [25] CVPR '23 | ViT-B | 69.8 | 70.5 | 41.3 |
| **ECA (Ours)** | **MiT-B1** | **70.9** | **71.4** | **42.9** |

**Table 1: Semantic segmentation results (mIoU%). $^\dagger$ denotes the methods use the saliency map as the auxiliary supervision. "BB.": Backbone. The backbone in multi-stage methods is that in the final segmentation model.**

its augmented training set (SBD) [14] that consists of 10582 images to train our ECA framework. MS COCO 2014 dataset contains training and validation sets with 81 classes, including 82,081 and 40,137 images respectively. Mean Intersection-Over-Union (mIoU) is reported as the evaluation metric for both CAM and final segmentation performance.

**Network Configuration.** In our work, we use Mix Transformer (MiT-B1) proposed in SegFormer [31] as our Transformer backbone, which shows the high training efficiency compared to the vanilla ViT model [37]. The segmentation head utilizes a MLP decoder that combines multi-level feature maps for prediction through simple MLP layers. The backbone is pretrained on ImageNet-1k, while other parameters are randomly initialized. For the DINO [5] model in the SAE module, we use its DeiT-S version with patch size = 8 to produce CARM by default.

**Implement Details.** During the training stage, we train ECA with an AdamW optimizer [20]. The initial learning rate is set as $6 \times 10^{-5}$ with a polynomial decay scheduler. The weight decay factor is set as 0.01. For the input images, they are cropped to $512 \times 512$, and applied augmentation strategies including random scaling with a range of [0.5, 2.0] and random horizontal flipping. The batch size is set as 8. The background threshold $\alpha$ is set to 0.5 by default. The other hyperparameter settings of PAR and auxiliary affinity learning loss (Section 3.4) are followed by [24].

| Method | BB. | val ($\mathcal{F}$) | val ($\mathcal{I}$) | ratio (%) |
|---|---|---|---|---|
| 1Stage [3] | WR38 | 80.8 | 62.7 | 77.6 |
| SLRNet [23] | WR38 | 80.8 | 67.2 | 83.2 |
| ToCo [25] | ViT-B | 80.5 | 69.8 | 86.7 |
| AFA [24] | MiT-B1 | 78.7 | 66.0 | 83.9 |
| TSCD [33] | MiT-B1 | 78.7 | 67.3 | 85.5 |
| **ECA (Ours)** | MiT-B1 | **78.7** | **70.9** | **90.1** |

Table 2: The segmentation performance of fully supervised counterparts. The results are evaluated on the VOC 2012 val set. The pixel-level ground-truths are directly used to supervise the segment decoder. $\mathcal{F}$: fully-supervised supervision. $\mathcal{I}$: image-level supervision (WSSS). *ratio*: val ($\mathcal{I}$) / val ($\mathcal{F}$).

For the experiments on PASCAL VOC 2012, we train the network for 18k iterations with 2k iterations warmed up for the classifiers. These images of the local view for the CCE module are cropped to $256 \times 256$. They are re-augmented by random color distortion and horizontal flipping. By default, the loss weights of $(\lambda_1, \lambda_2, \lambda_3)$ are all set as 0.1, $k$ in Equation 3 is set as 0.3, and the affinity threshold $\gamma$ in Equation 6 is set as 0.15. For the experiments on MS COCO 2014, we train the network for 80k iterations. The loss weights of $(\lambda_1, \lambda_2, \lambda_3)$ are set as $(0.1, 0.05, 0.1)$ and $k$ is set as 0.15, while other settings remain the same. In the inference stage, following the common practice in WSSS, we adopt multi-scale inference and CRF post-processing [6] to obtain the final segmentation results.

## 4.2 Experimental Results

**Comparison to State-of-the-art.** We report the segmentation performance of ECA on PASCAL VOC 2012 and MS COCO 2014 in Table 1. ECA achieves 70.9% (**+1.1%**), 71.4% (**+0.9%**), and 42.9% (**+1.6%**) mIoU on PASCAL VOC 2012 val, test and COCO val set, respectively, outperforming previous single-stage methods. Moreover, ECA demonstrates competitive performance compared to multi-stage methods, while significantly reducing the training cost. Notably, on the more challenging MS COCO 2014 dataset, our single-stage ECA even outperforms many recent multi-stage WSSS methods, demonstrating its robustness in more complex scenarios.

**Fully-Supervised Counterparts.** The single-stage competitors presented in Table 1 adopt various backbones. For a fair comparison, we examine the performance of their fully-supervised counterparts on the VOC 2012 val set. The results in Table 2 demonstrate that ECA significantly boosts the utilization of image-level annotations for semantic segmentation. ECA improves by **+4.6%** compared to methods using the same MiT-B1 backbone and by **+3.4%** compared to recent state-of-the-art work, *i.e.*, ToCo [25].

**Per-category segmentation performance.** The per-category segmentation results on PASCAL VOC val set is tabulated in Table 3. We can see the proposed ECA achieves remarkable performance improvement in most semantic classes (**16 of 21**). We can see the proposed ECA outperforms prior-arts over 5% in many classes. Especially, in the class of aeroplane, bicycle, cat, train, which are susceptible to the over-activation issue, ECA improve their performance by **+7.3%**, **+7.2%**, **+6.6%**, **+8.4%**, respectively.

| | RRM | 1Stage | AA&LR | AFA | **ECA (Ours)** |
|---|---|---|---|---|---|
| **bkg** | 87.9 | 88.7 | 88.4 | 89.9 | **92.2** |
| **aero** | 75.9 | 70.4 | 76.3 | 79.5 | **86.8** |
| **bicycle** | 31.7 | 35.1 | 33.8 | 31.2 | **42.3** |
| **bird** | 78.3 | 75.7 | 79.9 | **80.7** | 72.8 |
| **boat** | 54.6 | 51.9 | 34.2 | 67.2 | **73.4** |
| **bottle** | 62.2 | 65.8 | 68.2 | 61.9 | **72.7** |
| **bus** | 80.5 | 71.9 | 75.8 | **81.4** | 78.6 |
| **car** | 73.7 | 64.2 | **74.8** | 65.4 | 71.9 |
| **cat** | 71.2 | 81.1 | 82.0 | 82.3 | **88.9** |
| **chair** | 30.5 | 30.8 | 31.8 | 28.7 | **31.9** |
| **cow** | 67.4 | 73.3 | 68.7 | 83.4 | **83.8** |
| **table** | 40.9 | 28.1 | **47.4** | 41.6 | 41.1 |
| **dog** | 71.8 | 81.6 | 79.1 | 82.2 | **82.6** |
| **horse** | 66.2 | 69.1 | 68.5 | 75.9 | **81.5** |
| **motor** | 70.3 | 62.6 | 71.4 | 70.2 | **74.4** |
| **person** | 72.6 | 74.8 | **80.0** | 69.4 | 76.8 |
| **plant** | 49.0 | 48.6 | 50.3 | 53.0 | **58.3** |
| **sheep** | 70.7 | 71.0 | 76.5 | 85.9 | **86.6** |
| **sofa** | 38.4 | 40.1 | 43.0 | 44.1 | **47.5** |
| **train** | 62.7 | 68.5 | 55.5 | 64.2 | **76.9** |
| **tv** | 58.4 | **64.3** | 58.5 | 50.9 | 55.8 |
| **mIOU (%)** | 62.6 | 62.7 | 63.9 | 66.0 | **70.4** |

Table 3: Evaluation of per-category semantic segmentation results in mIoU on the val set of PASCAL VOC 2012. We compare the single-stage WSSS prior arts, including RRM [35], 1Stage [3], AA&LR [36], and AFA baseline [24]. The improvement over 5% are marked in color .

**Qualitative Results.** In Figure 5, we visualize the segmentation results of our ECA on the VOC 2012 val and MS COCO 2014 val sets, and compare them with the recent state-of-the-art methods, *i.e.*, AFA [24] and ToCo [25]. We can clearly observe that the proposed ECA demonstrates a significant improvement over AFA and ToCo. ECA effectively produces segmentation results that align more precisely with object boundaries, which are close to the ground-truths. Particularly, we can find that both AFA and ToCo produce many false-positive predictions caused by the pseudo-labels suffering from the semantic ambiguity problem of CAM. With the DINO as the semantic guider for segmentation, ECA can overcome this problem well. Also, we present the CAM visualization in Figure 5. We can observe that ToCo activates more non-target background regions and fails to activate some less discriminative regions. In contrast, the proposed ECA can achieve better object activations and provide high-quality CAM for seed selection and pseudo-label generation.

## 4.3 Ablation Studies and Analysis

**Ablation.** We investigate the impact of each component in our proposed ECA. The results of the ablation studies are reported in Table 4. It shows that our baseline setting achieves 62.2% and 63.5% mIoU of CAM and segmentation results on the PASCAL VOC val set. When introducing CARM to Affinity Learning (AL) [24], it

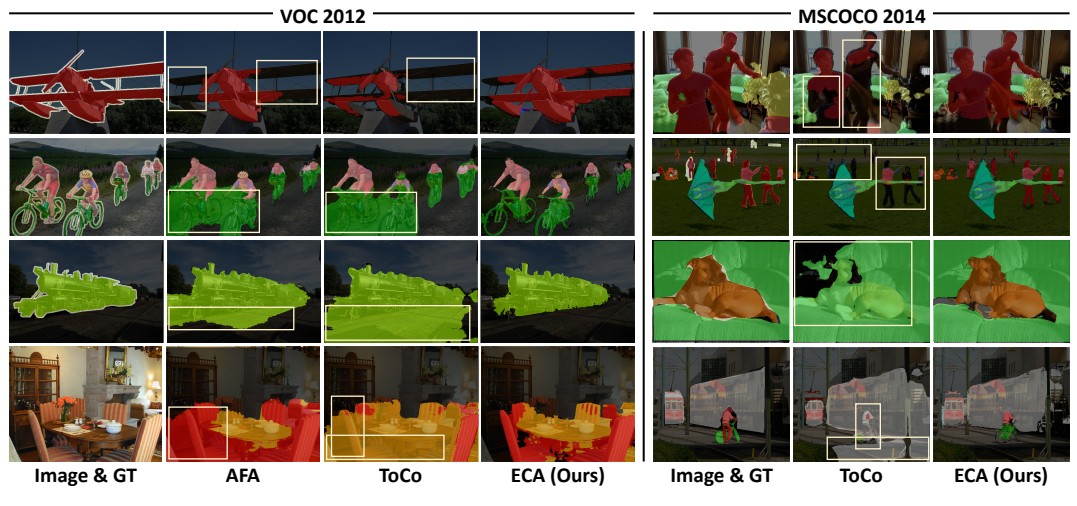

**Figure 4: Qualitative segmentation results comparison. From left to right, the results are predicted by AFA [24], ToCo [25] and the proposed ECA. Prediction results from AFA and ToCo produce some false positive/negative results. With CARM as the complementary guidance, ECA can solve this shortcomings well.**

| Method | AL | $\mathcal{L}_{seg}$ | $\tilde{\mathcal{L}}_{seg}$ | CCE | *CAM* | *Seg (msc)* |
|--------|----|----|----|-----|-------|-------------|
| Baseline | ✓[†] | ✓ | | | 62.2 | 63.5 |
| | ✓ | ✓ | | | 63.2 | 65.0 |
| | ✓ | | ✓ | | 66.6 | 67.1 |
| | ✓ | ✓ | ✓ | | 66.8 | 68.4 |
| **ECA** | ✓ | ✓ | ✓ | ✓ | **69.1** | **69.4** |

**Table 4: Ablation studies of ECA on the VOC 2012 val.** [†] **denotes Affinity Learning (AL) is supervised by CAM pseudo-labels. "*CAM*" is the CAM performance and "*Seg (msc)*" is segmentation performance with multi-scale inference. CRF processing is *not* implemented in the ablation study.**

improves the CAM and segmentation performance to 63.2% and 65.0%, respectively. Next, we validate the effectiveness of joint supervision (*i.e.*, $\mathcal{L}_{seg}$ and $\tilde{\mathcal{L}}_{seg}$) for the segmentation decoder. By replacing CAM with CARM to generate pseudo-labels as guidance, we observe improvements of 3.4% and 2.1% in CAM and segmentation performance, respectively. When we jointly use the CAM and CARM pseudo-labels, they can further improve the segmentation performance to 68.4%. We speculate that CAM and CARM serve complementary roles, where erroneous gradient updates caused by one pseudo-label are calibrated by the other, thereby enhancing the robustness and accuracy of the segmentation model. Furthermore, the CCE module significantly enhances the CAM performance with an increase of 2.3%, demonstrating that the local-to-global CAM correspondence benefits the robustness of CAM performance. With better CAM, which generates better pseudo-labels and seeds in SAE module, the segmentation performance further promotes to 69.4%, leading to the state-of-the-art.

**ViT Architecture in SAE.** The motivation of the SAE module is using the powerful class-agnostic affinity from DINO to yield

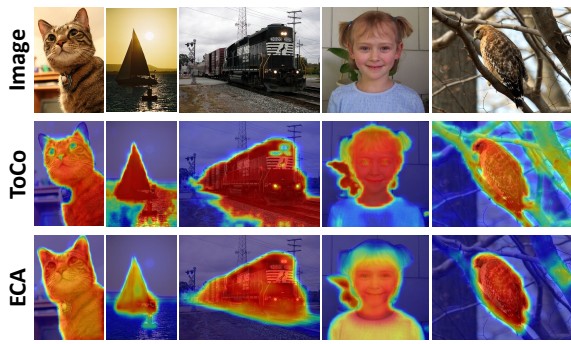

**Figure 5: Visualization of CAM. We compare the state-of-the-art one-stage approach, ToCo, with our proposed ECA. With DINO as the guidance, ECA can effectively suppresses the over-activation issue of CAM.**

complementary semantic guidance for WSSS, *i.e.*, CARM, through the CAM class and location priors. Different DINO variants influence the modeling of class-agnostic affinity, thus affecting the effectiveness of CARM. Here, we study the performance of ECA under different ViT models in Table 5. We can observe that the ViT fully supervised on ImageNet-1k cannot generate effective CARMs for semantic guidance, leading to catastrophic segmentation performance. Compared with other self-supervised ViT models [7, 22], we can find that DINO is the best choice for SAE. Further, we test the patch size of DINO and its ViT architecture. The results show that a small patch size of 8 is suitable for producing CARM, archiving the best segmentation performance, while a larger architecture (*i.e.*, ViT-B) would not benefit our proposed ECA.

**Analysis of CARM.** In Figure 6a, we visualize the effect of CARM calibration in SAE. We show some challenging cases where the initial seeds involve erroneous localizations, subsequently leading to inferior pseudo-labels. After CARM calibration, we can find that

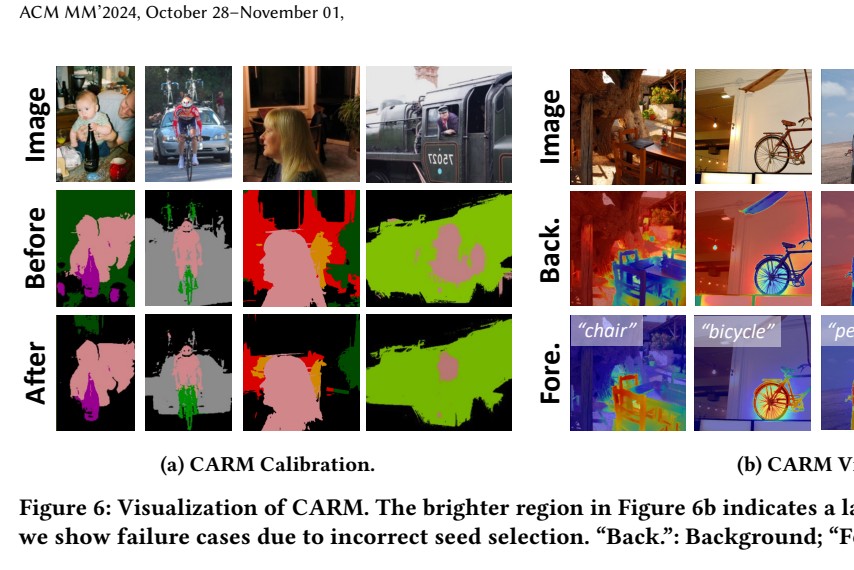

(a) CARM Calibration.

(b) CARM Visualization (after calibration).

Figure 6: Visualization of CARM. The brighter region in Figure 6b indicates a larger value. In the last two columns of Figure 6b, we show failure cases due to incorrect seed selection. "Back.": Background; "Fore.": Foreground.

| Model | Sup. | DINO | | | DINO v2 | MoCo v3 |
|---|---|---|---|---|---|---|
| | V-S/16 | V-S/16 | V-S/8 | V-B/8 | V-S/14 | V-S/16 |
| CAM | 43.7 | 68.4 | **69.1** | 69.3 | 69.5 | 66.4 |
| Seg | 36.7 | 67.2 | **69.4** | 69.3 | 68.6 | 65.7 |

Table 5: The performance comparison of different DINO versions and other self-supervised foundation models. "V-S" and "V-B" correspond to ViT-S and ViT-B backbones, while the value following ViT backbone indicates different patch size; "Sup.": ViT model fully supervised on ImageNet-1k. We also test the performance using DINO v2 [22] and MoCo v3 [7].

| top $k$% | Seg | Seg (msc) | Size | CAM | Seg (msc) |
|---|---|---|---|---|---|
| 5 | 67.0 | 69.4 | 96 | 66.4 | 67.5 |
| 10 | 67.1 | 69.5 | 128 | 67.6 | 68.8 |
| **30** | **67.6** | **69.4** | **256** | **69.1** | **69.4** |
| 50 | 66.8 | 68.7 | 384 | 69.2 | 69.5 |

(a) Top $k$% in seed selection.

(b) Crop size of local views.

Table 6: Impact of hyper-parameters. The performance is evaluated on PASCAL VOC 2012 `val` set.

numerous noise labels are successfully calibrated, thus improving the quality of these pseudo-labels. In Figure 6b, we demonstrate that CARM (after calibration) adeptly captures the foreground and background object information, enabling clear identification of the foreground objects. Notably, it's intriguing to observe that the background CARM exhibits strong cues of foreground objects. In single-class cases, the background CARM alone is competent in providing high-quality pseudo-labels for the segmentation decoder.

**Initial Seed Selection.** We utilize top $k$ activations of CAM as the initial seeds to exploit the class-aware affinity region for each labeled class. SAE with a smaller $k$ select fewer but more reliable seeds. Conversely, SAE with a larger $k$ select more seeds that cover broader object regions, thus benefiting the completeness of CARM. Table 6a shows $k = 0.3$ is the best choice but other values smaller than 0.3 can also yield favorable performance.

**Size of Local View.** The CCE module aligns CAM activations between global and local views to enhance the performance and reliability of CAM for better seed selection and CAM pseudo-labels. In Table 6b, we report the impact of local view size in CCE. It shows that the crop size of 256 and 384 both achieve favorable performance. Considering the computational load of establishing dense correlation pairs in CCE, we choose the crop size of 256 in our experiments by default.

## 5 CONCLUSION

This paper explores the distinct property of DINO to extract semantic guidance for single-stage WSSS, aiming to overcome the limitation of using CAM pseudo-labels. Specifically, highly-activated patches are selected as the initial seeds to induce Class-aware Affinity Region Map for each labeled class through the class-agnostic affinity map established by DINO. The CARM captures precise class-specific region affinities, thus offering high-fidelity pseudo-labels for both the segmentation decoder and affinity learning processes, enhancing the final segmentation outcomes. Moreover, motivated by the inconsistency between the global and local views, we propose CAM Correspondence Enhancement to align the CAM of global and local views through the dense correspondences, improving the performance of CAM for seed selection and the CAM pseudo-label generation. On PASCAL VOC 2012 and MS COCO 2014 datasets, our method achieves new state-of-the-art performance in single-stage WSSS and even outperforms numerous multi-stage WSSS methodologies on MS COCO 2014. From a broader view, the proposed method provides a novel perspective that leverages the powerful affinity modeling capability of DINO to excavate semantic guidance to overcome the CAM limitations.

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
