# OpenReview forum: "DINO is Also a Semantic Guider: Exploiting Class-aware Affinity for Weakly Supervised Semantic Segmentation"
_acmmm.org/ACMMM/2024/Conference — MM2024 Poster_

### Official Review · Reviewer_yR6V · 2024-05-12

**Rating:** 4
**Confidence:** 3

**Summary:**

The manuscript exploits the potential of semantic affinity existing in DINO and proposes a CAM Correspondence Enhancement module (CCE) to promote the dense local-to-global correspondences of CAM.

**Strengths:**

1. The manuscript introduces the DINO foundation model to CAM optimization.
2. The CAM is calibrated by the prototype method.
3. The local view is introduced to improve the performance.

**Limitations:**

The problems include:
1. “Foundation Model Assisted Weakly Supervised Semantic Segmentation”[1] has been proposed. The idea of using large model’s class-agnostic affinity graph is incremental.
2. Please discuss the computation cost of the proposed method.
3. The performance is worse than the reference [2], about 72.7 and 72.8.
4.  CAE in Fig 2 is error.
5. “Qualitative Results. In Figure 5” in page 6 should be Figure 4.

References:
[1] Yang X, Gong X. Foundation model assisted weakly supervised semantic segmentation[C]//Proceedings of the IEEE/CVF Winter Conference on Applications of Computer Vision. 2024: 523-532.
[2] Chen T, Yao Y, Huang X, et al. Spatial Structure Constraints for Weakly Supervised Semantic Segmentation[J]. IEEE Transactions on Image Processing, 2024.

**Suitability:**

2

---

### Official Review · Reviewer_zzi6 · 2024-05-21

**Rating:** 3
**Confidence:** 3

**Summary:**

This paper tackles the problem of weakly supervised semantic segmentation. A single-stage framework named ECA is proposed to fully exploit the potential of semantic affinity in DINO. Furthermore, a CCE module is designed to promote the dense local-to-global correspondences of CAM. Experimental results on benchmark datasets show that the proposed method outperforms previous methods.

**Strengths:**

1. This paper innovatively explores the utilization of DINO as a Semantic Guider for weakly supervised semantic segmentation.
2. This paper demonstrates improved performance compared to existing one-stage methods.
3. The rationale behind each of the new components (e.g., SAE, CCE) is intriguing.
4. The experiments and ablation studies are comprehensive and provide strong support for the author's claims.

**Limitations:**

1. The abstract involves multiple modules, but they lack clear logical connections, making it difficult to read.
2. The methods section lacks an overview of the entire framework, including details on the overall data flow.
3. The loss function L_{align} in Figures 2 and 3 is not introduced in the main text and is not part of the overall optimization objective in Equation 12. What does the red text "CAE" in Figure 2 represent?
4. The notation needs to be consistent throughout the text. For example, should there be a multiplication sign between the spatial dimensions HW? What is the relationship between N and F in line 228?
5. Why are the results for ECA in Tables 3 and 1 inconsistent while the results for other methods are consistent?
6. Are the results for ToCo in Table 1 consistent with their published results? They reported better results using ImageNet pre-training. Can the proposed ECA also benefit from the pre-training dataset used by ToCo?
7. Compared to the two-stage approach, the single-stage approach has the advantage of training efficiency. However, does it have an advantage in terms of training time complexity compared to other single-stage methods?
8. In Equation 12, the hyperparameters for the loss functions are set very low, making L_{cls}​ significantly dominant. What is the reason for this? Does simply assigning it a smaller weight affect performance?

**Suitability:**

2

---

### Official Review · Reviewer_rHc1 · 2024-05-24

**Rating:** 4
**Confidence:** 3

**Summary:**

This paper introduces a new approach, dubbed ECA, to Exploit the self-supervised Vision Transformer, DINO, inducing the Class-aware semantic Affinity to overcome this limitation. Specifically, It introduces a Semantic Affinity Exploitation module (SAE). It establishes the class-agnostic affinity graph through the self-attention of DINO. Using the highly activated patches on CAMs as “seeds”, it propagates them across the affinity graph and yield the Class-aware Affinity Region Map (CARM) as supplementary semantic guidance. Experimental results demonstrate that ECA improves the model's object pattern understanding.

**Strengths:**

* Novelty: The ECA framework introduces a novel Semantic Affinity Exploitation (SAE) module that leverages the self-supervised Vision Transformer DINO to generate Class-aware Affinity Region Maps (CARM) as complementary semantic guidance. This innovative approach effectively addresses the limitations of traditional Class Activation Maps (CAMs) in WSSS.
* Theoretical Approach: By incorporating DINO to induce class-aware affinity from CAMs, ECA bridges the gap between class and localization tasks, providing more accurate pseudo-labels for segmentation supervision. This theoretical approach enhances the quality of segmentation outcomes and overcomes the ambiguity inherent in CAMs.
* Technical Correctness: The ECA framework demonstrates technical correctness by effectively utilizing the semantic properties of DINO to guide the segmentation process. The proposed SAE module and CAM Correspondence Enhancement (CCE) module contribute to improving the quality of pseudo-labels and segmentation results, showcasing the technical soundness of the approach.
* Adequate Evaluation: The paper provides thorough empirical evidence and ablation studies to validate the effectiveness of the ECA framework. Comparative results with state-of-the-art methods such as AFA and ToCo demonstrate significant improvements in segmentation accuracy and object boundary alignment. The evaluation methodology ensures the robustness and reliability of the proposed approach.
*  Clarity: The paper effectively communicates the proposed framework, modules, and experimental results in a clear and concise manner. The methodology, results, and implications are well-explained, making it easy for readers to understand the contributions and significance of the ECA framework.
* Applications: The ECA framework has practical applications in various domains requiring semantic segmentation, especially when dealing with limited annotation resources. The improved performance in single-stage WSSS using image-level labels opens up possibilities for more efficient and accurate segmentation tasks in real-world applications.

**Limitations:**

* While the proposed ECA framework introduces innovative modules like Semantic Affinity Exploitation (SAE) and CAM Correspondence Enhancement (CCE) to enhance segmentation performance, the fundamental concept of leveraging semantic affinity from DINO for segmentation guidance has been explored in prior works.
* The paper could benefit from a more extensive evaluation to validate the generalizability and robustness of the proposed ECA framework 2. While the empirical results on the PASCAL VOC dataset demonstrate significant improvements in per-category segmentation performance, a broader evaluation on diverse datasets or real-world scenarios would strengthen the paper's contributions and practical applicability.

**Suitability:**

2

---

### Official Review · Reviewer_ZJV4 · 2024-05-28

**Rating:** 3
**Confidence:** 4

**Summary:**

The article proposes a novel approach to generate semantic guidance using the self-supervised nature of DINO model to address the limitations of CAM pseudo-tagging when using only image-level tags for semantic segmentation. Specifically, this paper proposes semantic affinity exploitation module to establish class-agnostic affinity graph by self-attention of DINO. Further, the class-aware affinity region map is yielded as supplementary semantic guidance. The paper also proposes CCE to encourage dense local-to-global CAM correspondences.

Experiments are conducted on VOC and COCO to validate the proposed methods.

**Strengths:**

1. The article designs a Semantic Affinity Exploitation module that can efficiently utilize the semantic affinity of DINO to generate high-quality Class-Aware Affinity Region Mapping (CARM), thus providing high-fidelity pseudo-labels for the segmentation decoder and affinity learning process.

2. The article proposes a CAM Correspondence Enhancement module that can improve the performance of CAM by establishing a dense correspondence between global and local perspectives, thus improving seed selection and CAM pseudo-label generation.

3. New state-of-the-art performance was achieved for the article's approach on both PASCAL VOC and MS COCO datasets.

**Limitations:**

1. There is an insufficient introduction of DINO, which is important in the paper.

2. ECA does not go far enough in explaining the workings of the DINO model and the limitations of CAM and could have further elaborated on the connection and motivation for combining the two.

3. The analysis of the method's time complexity and computational resource consumption is not comprehensive enough, and consideration could be given to include an analysis and discussion of this aspect.

**Suitability:**

2

---

### Meta-Review · Area_Chair_MLoe · 2024-06-29

**Recommendation:** Accept (Poster)
**Confidence:** 4

**Metareview:**

The contributions offer reasonable novelty. State-of-the-art results.